# Few-Shot Fast-Adaptive Anomaly Detection

**Ze Wang** [*][†]**, Yipin Zhou**[‡]**, Rui Wang**[‡]**, Tsung-Yu Lin**[‡]**, Ashish Shah**[‡]**, and Ser-Nam Lim**[‡]
[†]Purdue University      [‡]Meta AI

## Abstract

The ability to detect anomaly has long been recognized as an inherent human ability, yet to date, practical AI solutions to mimic such capability have been lacking. This lack of progress can be attributed to several factors. To begin with, the distribution of "abnormalities" is intractable. Anything outside of a given normal population is by definition an anomaly. This explains why a large volume of work in this area has been dedicated to modeling the normal distribution of a given task followed by detecting deviations from it. This direction is however unsatisfying as it would require modeling the normal distribution of every task that comes along, which includes tedious data collection. In this paper, we report our work aiming to handle these issues. To deal with the intractability of abnormal distribution, we leverage Energy Based Model (EBM). EBMs learn to associate low energies to correct values and higher energies to incorrect values. At its core, the EBM employs Langevin Dynamics (LD) in generating these incorrect samples based on an iterative optimization procedure, alleviating the intractable problem of modeling the world of anomalies. Then, in order to avoid training an anomaly detector for every task, we utilize an adaptive sparse coding layer. Our intention is to design a plug and play feature that can be used to quickly update what is normal during inference time. Lastly, to avoid tedious data collection, this mentioned update of the sparse coding layer needs to be achievable with just a few shots. Here, we employ a meta learning scheme that simulates such a few shot setting during training. We support our findings with strong empirical evidence.

## 1 Introduction

Anomaly detection is an important area of study in the field of artificial intelligence. It has found utility in computer vision applications such as industrial inspection [6] and video surveillance [28, 61, 39], in the context of abuse prevention such as misinformation, fraud and network intrusion detection [60, 8, 35], and others such as system health monitoring and fault detection [4, 42]. In this paper, we propose an approach for detecting anomaly in images, where we have carefully designed steps to handle some of the bigger issues that have prevented the deployment of image anomaly detection in the real-world.

Image anomaly detection can generally be defined as the identification of abnormalities in a given image. An exact definition of abnormality in this case is elusive because abnormality can be derived from any unknown distribution outside of a normal population. Many studies have hence focused on modeling the normal population instead of learning irregularities, where the goal is to capture the shared concept among all of the normal data as one or several reference models. This process usually requires investing significant efforts in curating a large set of normal samples for *each task*, after which anomaly is detected as deviations from the reference model(s) [1, 58]. Recent work from [50] provides algorithms that utilize only a few normal samples to train models from scratch. However, the models still have to be provisioned for each new task, which requires considerable human efforts

---

[*]Work done as an intern at Meta AI. Contact: `zewang@purdue.edu`

36th Conference on Neural Information Processing Systems (NeurIPS 2022).

and expertise, and thus lack the fast deployment criterion that is often time critical for real-world applications. In view of these challenges, our goals for this work are threefold. We are interested in designing an anomaly detection system that is capable of: (G1) modeling the normal population while at the same time has a principled approach towards modeling the abnormalities; (G2) quickly adapting to a new task at inference time; and (G3) requiring only a few normal shots to update itself to the new task at hand.

For (G1), we introduce the class of Energy Based Model (EBM), which is an important family of generative models [62, 17, 57]. EBMs have been shown to demonstrate superior capability on modeling data density and localizing anomaly [20]. For our purpose, the EBM we adopted learns to assign low energy to normal samples but high energy to abnormal samples. More importantly, the abnormal samples are generated with a procedure known as Langevin Dynamics (LD) [54], which, in its original form, starts with a noise image and gradually samples from the distribution along the direction of lower energy. This lends itself gracefully to utilizing the generated intermediate samples as negative/abnormal. The LD procedure is then coupled with maximum likelihood loss [24] that aims to maximize the energy differences between the normal and abnormal samples.

To achieve (G2), we propose an adaptive sparse coding layer that is attached to the deep feature extractor in the EBM as Figure 1 shows. The extracted deep feature is forwarded to the sparse coding layer, where the dictionary is constructed with the features of a few normal samples of the given task. In essence, the input representation has been decomposed into a linear combination of normal features with the sparsity constraint imposed. The final energy score is measured by the distance between the original and the reconstructed features (after the sparse coding layer). Under this scheme, the dictionary for a particular task is not obtained by learning, but instead is constructed by the feature representations of a few normal samples during inference. As a result, this simple "plug-and-play" trick allows the model to be adapted to novel tasks promptly without re-training. Further, we expect that the dictionary, which is formed by normal features, will not be able to explain the abnormal samples well, causing relatively high reconstruction error that lends itself for subsequent detection. As a bonus, a backward pass of energy score minimization can be used for localizing abnormal regions. We show that using gradient to localize anomalies yields superior robustness.

Towards (G3), we utilize meta learning [52, 18] to simulate the scenario of being given a new task with a few normal shots to update the dictionary, followed by training the EBM. This is accomplished by episodic training, where in each episode the model is adapted to a held back task that is given a few normal samples. To accelerate the EBM training, we introduce "learning from inpainting", a simple yet effective strategy for synthesizing hard abnormal samples quicker by starting the LD procedure with a synthesized image that is simply a normal sample with a noise patch injected as opposed to a noise image that is traditionally what is used.

We show the proposed few-shot fast-adaptive anomaly detection and localization framework is able to efficiently adapt to a novel task (e.g., a new object category or scenes from a new camera) with a few normal samples without training on both industrial inspection and video surveillance. Compared with previous methods that adapts to new task through either from scratch training in few shots [50, 55] or few-shot with few steps of gradient descent [32], the proposed framework is the first that performs task adaptation with a single forward pass and without any gradient descent. Despite the fast adaptation, we provide both qualitative and quantitative results to demonstrate that our method outperforms other adaptive frameworks and is comparable to methods that rely on large amount of normal samples.

## 2 Backgrounds

In this section, we briefly introduce two key ingredients of the proposed method: EBMs and sparse coding.

**Energy-based Model.** In EBMs, the goal is to learn an energy function $E_\theta(\mathbf{x}) : \mathbb{R}^d \to \mathbb{R}$ which parametrizes the data density $p_\theta(\mathbf{x})$ as:

$$p_\theta(\mathbf{x}) = \frac{\exp(-E_\theta(\mathbf{x}))}{\int_\mathbf{x} \exp(-E_\theta(\mathbf{x}))}, \tag{1}$$

where $\theta$ is the parameter of the energy function and $Z_\theta = \int_\mathbf{x} \exp(-E_\theta(\mathbf{x}))$ is the partition function. Approximating the true data distribution $p_{\text{data}}(\mathbf{x})$ is equivalent to minimizing the expected negative

log-likelihood function over the data distribution, defined by the loss function:

$$\mathcal{L}_{\text{ML}} = \mathbb{E}_{\mathbf{x} \sim p_{\text{data}}(\mathbf{x})}[-\log p_\theta(\mathbf{x})] = \mathbb{E}_{\mathbf{x} \sim p_{\text{data}}(\mathbf{x})}[E_\theta(\mathbf{x}) + \log Z_\theta]. \tag{2}$$

As the computation of $\mathcal{L}_{\text{ML}}$ involves an intractable term $Z_\theta$, the common practice is to represent the gradient of $\mathcal{L}_{\text{ML}}$ as,

$$\nabla_\theta \mathcal{L}_{\text{ML}} = \mathbb{E}_{\mathbf{x}^+ \sim p_{\text{data}}(\mathbf{x})}[\nabla_\theta E_\theta(\mathbf{x}^+)] - \mathbb{E}_{\mathbf{x}^- \sim p_\theta(\mathbf{x})}[\nabla_\theta E_\theta(\mathbf{x}^-)]. \tag{3}$$

This objective decreases the energy of positive data samples $\mathbf{x}^+$ from the true distribution (normal samples in our use case) and increases the energy of negative samples $\mathbf{x}^-$ from the model $p_\theta$ (synthesized abnormal samples). In practice, the synthesized negative samples are achieved through Langevin dynamics [54], which a $J$-steps sampling along the direction of energy minimization is given by:

$$\tilde{\mathbf{x}}^j = \tilde{\mathbf{x}}^{j-1} - \frac{\beta}{2} \nabla_{\mathbf{x}} E_\theta(\tilde{\mathbf{x}}^{j-1}) + \omega^k, \quad \omega^k \sim \mathcal{N}(0, \beta \mathbf{I}), \quad j = \{1, \ldots, J\} \tag{4}$$

where $\beta$ is the step size, and the initialization $\mathbf{x}^0$ is sampled from a predefined prior distribution. The synthesizing ability of EBMs enables generating abnormal samples to help in learning a more accurate data density, and is often touted as the one of the advantages of using an EBM.

**Sparse coding.** Approximating a signal $\mathbf{z} \in \mathbb{R}^d$ with the sparse linear combination over a dictionary $\mathbf{D} \in \mathbb{R}^{d \times k}$ can be expressed as:

$$\min_{\boldsymbol{\alpha}} \frac{1}{2} ||\mathbf{z} - \mathbf{D}\boldsymbol{\alpha}||_2^2 + \lambda ||\boldsymbol{\alpha}||_1, \tag{5}$$

where $\boldsymbol{\alpha}$ is the sparse coefficients, with its sparsity ($l_1$ norm) and $\lambda$ is the weight of the sparsity constraint. $\mathbf{D}\boldsymbol{\alpha}$ is a sparse approximation to the original signal $\mathbf{z}$. In practice, finding the dictionary atoms and the sparse coefficients is usually formulated as an optimization problem.

In this paper, we adopt Iterative Soft Thresholding Continuation (ISTC) [25] to convert this optimization problem into linear operations with a non-linear shrinkage function, which allows sparse coding to be seamlessly integrated into the deep neural networks. To compute a sparse coefficient $\boldsymbol{\alpha}$, ISTC performs iterations of gradient steps on reconstruction $||\mathbf{z} - \mathbf{D}\boldsymbol{\alpha}||^2$ and a proximal projection step to increase coefficient sparsity.

Formally, initializing the coefficients at the first step $\boldsymbol{\alpha}_0$ with all zeros, each step of ISTC refines the sparse code with descending values of $\lambda$ from $\lambda_{\text{max}}$ to $\lambda_\star$: each step of ISTC is expressed as:

$$\boldsymbol{\alpha}_{n+1} = \sigma(\boldsymbol{\alpha}_n + \mathbf{D}^\top(\mathbf{z} - \mathbf{D}\boldsymbol{\alpha}_n), \lambda_n), \quad \text{with} \quad \lambda_n = \lambda_{\text{max}} \frac{\lambda_{\text{max}}}{\lambda_\star}^{-n/N}, \tag{6}$$

where $\sigma(\cdot, \cdot)$ here is a shrinkage function that truncates small values (lower than $\lambda$) of the coefficients to 0 to enforce sparsity, and can be easily implemented by a customized ReLU activation function:

$$\sigma(\mathbf{z}, \lambda) = \text{sgn}(\mathbf{z})(\max(|\mathbf{z}| - \lambda, 0)) = \text{sgn}(\mathbf{z})\text{ReLU}(|\mathbf{z}| - \lambda). \tag{7}$$

# 3 Proposed Method

In this section, we describe the proposed fast adaptive anomaly detection framework in details. In Section 3.1, we introduce the adaptive EBM which consists of a deep feature extractor followed by an adaptive sparse coding layer. From there, we further show that utilizing larger receptive field in the sparse coding could improve training robustness (Section 3.1.1), and applying smoothed shrinkage functions could help speed up convergence (Section 3.1.2). In Section 3.2, we describe the episodic training regime on various anomaly detection tasks that mimics few-shot adaptation in the meta-testing stage while learning common knowledge across tasks. Instead of synthesizing negative samples (anomaly) directly from noise, we introduce a simple but effective "learning from inpainting" operation to accelerate the training in Section 3.3. Finally, we summarize training steps of the proposed method in Algorithm 1, and inference steps in Algorithm 2.

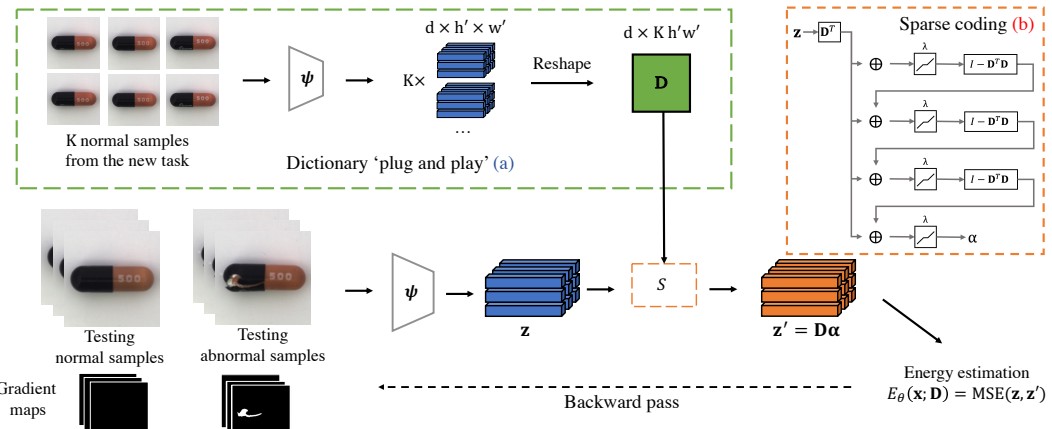

Figure 1: Overview of the inference stage on a new task. (a) Adapting the task-specific dictionary with K normal samples. (b) Three iterations of sparse coding based on Eqn. 6. We also show a backward pass from the reconstruction error to localize the abnormal regions

## 3.1 Adaptive Energy-based Model

An EBM is a form of generative model and it is widely used for modeling data density and sampling. While there has been recent work [20] applying EBM to anomaly detection, it still requires re-training for each new task. To efficiently adapt the EBM to novel tasks, we introduce an adaptive sparse coding layer which is conditioned on the dictionary constructed by the features of normal samples. Specifically, as illustrated in Fig 1, given an input image, $\mathbf{x} \in \mathbb{R}^{3 \times h \times w}$, we first obtain the corresponding feature $\mathbf{z} \in \mathbb{R}^{d \times h' \times w'}$ from the deep feature extractor $\mathbf{\Psi}$ with parameters $\theta$, so that $\mathbf{z} = \mathbf{\Psi}(\mathbf{x}; \theta)$. All feature vectors along spatial axes of $\mathbf{z}$ are then sparsely decomposed through the sparse coding layer over a task-specific dictionary $\mathbf{D} \in \mathbb{R}^{d \times Kh'w'}$, which contains the features of $K$ normal samples of the current task as shown in the Fig 1(a). Each feature vector of the normal sample features is then directly used as an atom in the task dictionary. The decomposed coefficients are $\alpha = \boldsymbol{\mathcal{S}}(\mathbf{z}; \mathbf{D})$, where $\alpha \in \mathbb{R}^{Kh'w' \times h' \times w'}$ and $\boldsymbol{\mathcal{S}}$ denotes the iterative sparse decomposition process of (6). By multiplying the coefficient $\alpha$ with the dictionary $\mathbf{D}$, we obtain the reconstructed features $\mathbf{z}' = \mathbf{D}\alpha$. The sparsity regularization to $\alpha$ is important, as it encourages input features to be reconstructed by simple combinations of dictionary atoms (normal features), so that it would be difficult for features of abnormal samples to be well-approximated, therefore producing higher reconstruction errors that make it conducive for detecting anomalies. From here, the final energy score is formulated as the mean squared error (MSE) between the original and the reconstructed features:

$$E_\theta(\mathbf{x}; \mathbf{D}) = \text{MSE}(\mathbf{z}, \mathbf{z}') = ||\mathbf{\Psi}(\mathbf{x}; \theta) - \mathbf{D}\boldsymbol{\mathcal{S}}(\mathbf{\Psi}(\mathbf{x}; \theta); \mathbf{D})||^2. \quad (8)$$

In effect, Eqn. 8 depicts a conditional EBM, which is conditioned on the task-specific $\mathbf{D}$ formed by normal features. With the energy score, we can obtain pixel-wise **anomaly localization maps** through $\nabla_{\mathbf{x}} - E_\theta(\mathbf{x}; \mathbf{D})$, i.e., the gradients of pixels along the direction of minimization. High gradient magnitudes indicate regions that cannot be well explained by the dictionary $\mathbf{D}$. Modifications to these regions can potentially remove the anomaly and reduce the energy as in Eqn. 4. In Section 4.1 and Appendix Section B.5, we show that using the gradient (as a natural ingredient of EBMs with LD) is more robust compared with auto-encoder and reconstruction based methods to generalize well to unseen tasks (Appendix Figure C). In the following sections, we will discuss how to make the training of this adaptive structure more robust.

### 3.1.1 Sparse Coding with Receptive Field.

As discussed in Section 3.1, the input feature $\mathbf{z}$ is represented as $h' \times w'$ of $d$-dim feature vectors and they are treated independently while passing through the sparse coding layer. The region of the input image that affects one feature vector is determined by the receptive field of the feature extractor. The trade-off is that a small receptive field may not capture enough contextual information, while applying a large receptive field would make feature maps spatially coarse and make it hard to

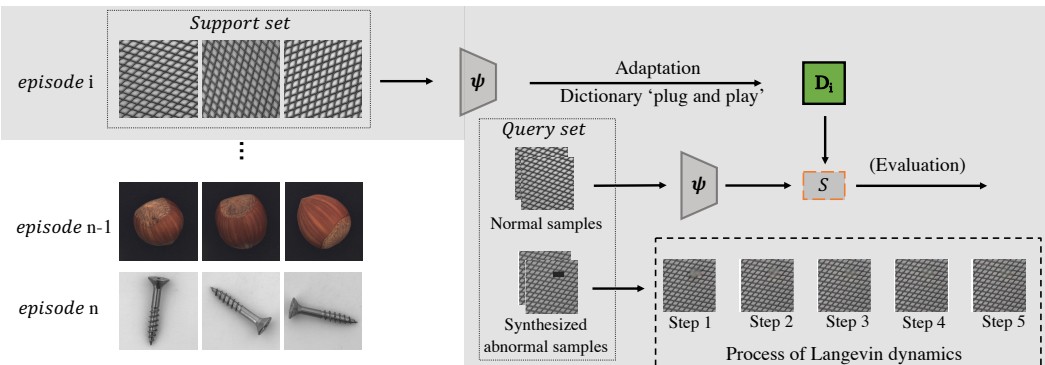

Figure 2: Illustration of episodic training and (a) "learning by inpainting". In each episode, a support set is constructed with normal samples. The features of the normal samples are plugged into the adaptive sparse coding layer as the dictionary. Synthesized abnormal samples are corrected by performing gradient corrections with the gradient obtained by the direction of energy minimization.

spot small anomaly regions. To solve this dilemma, instead of carefully tuning the receptive field of each layer of the feature extractor, we introduce a simple yet effective technique of applying the receptive field on the sparse coding layer. Specifically, as illustrated in Appendix Fig A, rather than performing sparse coding to each individual d-dim feature vectors, we apply it on $d \times l \times l$ volumes centered around each feature vector, where $l$ is the receptive field. This is equivalent to applying a $l \times l$, sliding window on spatial axes of the feature map and can be easily implemented by *image to column* (*Im2Col*) operation. Then we flatten the feature volumes into $dl^2$-dim vectors and adjust the shape of the dictionary accordingly. In this way, we are able to capture contextual information without needing to carefully tune the architecture of the feature extractor and we show in the later experiments that this technique improves the robustness of the network on different types of objects.

### 3.1.2 Shrinkage Function

The effectiveness of training the EBM for localizing anomaly regions heavily depends on the gradient propagation from later to earlier layers. It is shown in [15] that smooth activation functions like Swish [45] could be beneficial here. Notably, the gradients of the dictionary $\mathbf{D}$ are determined by the sparse coding coefficients $\alpha$ as shown in Eqn. 6. However, the sparsity constraint of $\alpha$ would turn off the gradient computation of many elements in $\mathbf{D}$ and this could be detrimental during the early stage of the training. To alleviate the sparse gradient issue, we replace the RELU-like shrinkage function in Eqn. 7 with its smoothed counterparts by introducing the Sigmoid based shrinkage functions (SigShrink). The SigShrink function is originally proposed for non-parametric signal estimation in [3], and can be defined as:

$$\sigma_\tau(\mathbf{z}, \lambda) = \frac{\mathbf{z}}{1 + \exp(-\tau(|\mathbf{z}| - \lambda))}, \tag{9}$$

where $\tau$ is the hyperparameter of smoothness. We present visualizations of the hard shrinkage function Eqn. 7 and SigShrink with different values of $\tau$ in Fig B. Comparing to the hard shrinkage function which truncates small values into zeros, the SigShrink with a large $\tau$ can sharply force small values to near-zeros. Therefore, the SigShrink will guarantee non-zero gradients everywhere.

### 3.2 Episodic Training

To train the proposed adaptive EBM, we perform episodic training that is widely adopted by meta-learning few-shot learners [18, 51]. Following the terminology of few-shot learning, in each training episode, the model is adapted and tested with a task sampled from the underlying task distribution. Specifically, the model is adapted to a support set of the given task, then a query set with ground truth labels is applied to evaluate the adaptation, which is used to update the model parameters. As shown in Fig 2, the support set of the $i$-th episode task contains a small number of $K$ normal samples $\{\mathbf{s}_k^i\}_{k=1}^K$. The features $\mathbf{z}_k^i = \mathbf{\Psi}(\mathbf{x}_k^i; \theta)$ of these normal samples are plugged into the dictionary $\mathbf{D}^i \in \mathbb{R}^{d \times Kh'w'}$ corresponding to the $i$-th task to adapt the dictionary. After that, the adapted model

is measured by a query set consisting of $M$ normal samples $\{\mathbf{q}_m^i\}_{m=1}^M$ and $M$ abnormal samples $\{\hat{\mathbf{q}}_m^i\}_{m=1}^M$. Note that there is no actual abnormal samples given during training, instead, they are iteratively sampled from the EBM and the sampling will be discussed in details in Section 3.3. Recall that the training of EBM with contrastive divergence as in Eqn. 3 requires the estimation of energy scores of both positive samples from the true data distribution and negative samples from the modeled distribution. The positive energy can be estimated empirically with normal query set samples. The negative energy can be estimated by performing the MCMC-based sampling technique [37, 54], typically Langevin Dynamics as described in Eqn. 4. Denoting the output of Langevin dynamics (sampled abnormal samples) initialized with $\hat{\mathbf{q}}_m^i$ as $\mathbf{LD}(\hat{\mathbf{q}}_m^i)$, we have the empirical estimation of the contrastive divergence of the $i$-th episode as:

$$\mathcal{L}_{\text{cd}} = \frac{1}{m} \sum_{m=1}^M \left[ E_\theta(\mathbf{q}_m^i; \mathbf{D}^i) - E_\theta(\mathbf{LD}(\hat{\mathbf{q}}_m^i); \mathbf{D}^i) \right]. \tag{10}$$

With the energy score equivalent to the feature reconstruction error in Eqn. 8, minimizing $\mathcal{L}_{\text{cd}}$ encourages normal features to be well-reconstructed by a sparse linear combination of dictionary atoms while the features from abnormal samples tend to produce relatively higher reconstruction errors so that they can be easily spotted.

### 3.3 Synthesizing Negative Samples

Typical EBM training with contrastive divergence conducts negative sampling from the modeled density using techniques such as Langevin Dynamics, which applies gradient descent to a noise initialization with small step size and large number of steps [17]. Such negative sampling steps can be costly and we argue that it is unnecessary in our case. Instead, we introduce a new strategy of "learning by inpainting". Starting from a positive query sample $\mathbf{q}_m^i$, we synthesize the corresponding negative sample $\hat{\mathbf{q}}_m^i$ by randomly placing a small uniform noise patch on the image. The Langevin Dynamics procedure is then initialized with the resulting image instead of a noise image. As the Langevin Dynamics proceeds, synthesized abnormal samples $\mathbf{LD}(\hat{\mathbf{q}}_m^i)$ are inpainted along the direction of "normal", $\mathbf{q}_m^i$, and we introduce the following reconstruction loss:

$$\mathcal{L}_{\text{rec}} = \frac{1}{m} \sum_{m=1}^M \text{MSE}(\mathbf{LD}(\hat{\mathbf{q}}_m^i), \mathbf{q}_m^i). \tag{11}$$

We show in Fig 2(a) that, starting from a synthesized abnormal sample, only 5 steps of Langevin dynamic would be sufficient to make it visually close to the corresponding normal sample during training, serving as "hard negatives" that further facilitates the learning. The final loss of the episodic training is simply:

$$\mathcal{L} = \eta_0 \mathcal{L}_{\text{rec}} + \eta_1 \mathcal{L}_{\text{cd}}, \tag{12}$$

where $\eta_0$ and $\eta_1$ are balance two loss terms. We summarize the overall training and inference procedure in Alg. 1 and Alg. 2 respectively.

## 4 Experiments

In this section, we conduct evaluation on the industrial inspection task with the MVTec-AD dataset [5, 6] (Section 4.1). Even though our proposed framework is image-based, we further demonstrate it's efficacy on the video anomaly detection task in Section 4.2. In Section 4.3, we show ablations and insights relating to the adaptive sparse coding components. We show additional ablations including the superiority of using gradient of EBMs over pixel-wise reconstruction to localize anomalies in App. B and we provide implementation details in App. A.

### 4.1 Industrial Inspection

The goal of this anomaly detection task is to predict whether a manufactured component contains any defects. The MVTec-AD dataset includes 15 categories of object. To demonstrate the fast adaptation capability of the proposed method, we adopt a *leave-one-out* training strategy. Specifically, samples of each target category are reserved for testing only, and the episodic training is performed on the

---

**Algorithm 1** Training procedure.

---

1: **Given**: A feature extractor $\mathbf{\Psi}$ with parameter $\theta$; a training dataset of multiple tasks with positive (normal) samples only.
2: **Given**: Number of shots $K$; number of query samples $Q$; step size $\beta$ of Langevin dynamics; total training episodes $I$; and learning rate $\epsilon$.
3: Initialize the feature extractor $\mathbf{\Psi}$.
4: **for** Episode $i = 1 : I$ **do**
5:     Sample the $i$-th task from the dataset, and randomly pick $K + M$ samples to form the support set $\{\mathbf{s}_k^i\}_{k=1}^K$ and the query set $\{\mathbf{q}_m^i\}_{m=1}^M$.
6:     Generate corrupted query samples $\{\hat{\mathbf{q}}_m^i\}_{m=1}^M$ by placing random patches to $\{\mathbf{q}_m^i\}_{m=1}^M$.
7:     Extract the support and query sample features with $\mathbf{\Psi}$ and update the adaptive sparse coding layer with $i$-th task dictionary $\mathbf{D}^i$, which is constructed by support sample features. The energy function of the $i$-th task is now parametrized by $E_\theta(\cdot, \mathbf{D}^i)$.
8:     Obtain synthesized negative samples $\{\mathbf{LD}(\hat{\mathbf{q}}_m^i)\}_{m=1}^M$ with the updated energy function using Langevin dynamic in (4).
9:     Obtain the final loss $\mathcal{L}$ with $\mathcal{L}_{\text{cd}}$ (10) and $\mathcal{L}_{\text{rec}}$ (11).
10:     Update parameters $\theta \leftarrow \theta - \epsilon \nabla_\theta \mathcal{L}$.
11: **end for**
12: **Return** $\mathbf{\Psi}$ with parameter $\theta$.

---

**Algorithm 2** Inference procedure on a task indexed by $i$.

---

1: **Given**: Feature extractor $\mathbf{\Psi}$ with trained parameter $\theta$.
2: **Given**: The support set $\{\mathbf{s}_k^i\}_{k=1}^K$ and query samples $\{\mathbf{q}_m^i\}_{m=1}^M$ to be tested.
3: Extract normal feature tensor $\mathbf{Z}^i \in \mathbb{R}^{K \times d \times h' \times w'}$.
4: Reshape the normal feature tensor into a matrix, and use it as the task-specific dictionary $\mathbf{D}^i \in \mathbb{R}^{d \times Kh'w'}$.
5: Estimate the anomaly score of a test sample $\mathbf{q}_m^i$ by its energy score using Eqn 8, with $E_\theta(\mathbf{q}_m^i; \mathbf{D}^i) = ||\mathbf{\Psi}(\mathbf{q}_m^i; \theta) - \mathbf{D}^i \mathcal{S}(\mathbf{\Psi}(\mathbf{q}_m^i; \theta); \mathbf{D}^i)||^2$, where a higher energy score indicates that $\mathbf{q}_m^i$ is more likely to be an abnormal sample.
6: The pixel-wise anomaly map of a test sample $\mathbf{q}_m^i$ can be obtain by visualizing $\nabla_{\mathbf{q}_m^i} - E_\theta(\mathbf{q}_m^i; \mathbf{D}^i)$, where abnormal regions show higher magnitude.

---

remaining categories. During the training stage, the model will not see any samples from the target category. During testing, we first adapt the model to the target category with *10 randomly selected normal samples*, then measure the performance with the entire testing set. We run the test 5 times, each time the model is adapted to random sets of 10 normal samples from the target category.

In Table 1, we first show performance of "upper-bound" methods, which train each category from scratch with massive normal samples. Specifically, [7, 6] train auto-encoders (AE) with normal samples and measure the reconstruction errors during the inference; AnoGAN [48] adopts a generative adversarial network (GAN) to learn the manifold of normal; VE-VAE [29] presents a visually explainable variational auto-encoder through gradient-based attention. For apple-to-apple comparison, we create a strong baseline by applying model-agnostic meta-learning [18] on an AE (denoted as MAML-AE, more details in App. Sec. A.3). To the best of our knowledge, the proposed method is the first that is capable of producing strong performance on new tasks, using **just a single forward pass and no further training**. This strongly suggests that the learned parameters are effectively shared across tasks, greatly helping to accelerate the model deployment process that is typically cumbersome otherwise. All results from our methods are obtained with flipping and $90^o$ rotation as augmentations. The proposed method outperforms MAML-AE by a large margin. Our results are even competitive with the "upper-bounds" in some categories. We show the localized anomaly regions from our method in Fig 3. Additional visualizations are in the App. Fig D.

## 4.2 Video Surveillance

In video anomaly detection, a common goal is to detect abnormal events captured by surveillance cameras (e.g., a motorcycle on the sidewalk). A model trained on videos from one camera might not generalize well on other cameras due to different locations / mounting heights / lightning conditions, and it is not feasible to train one model for every new camera in practice. The ability to quickly adapt to new scenes is a significant contribution to the task of video surveillance. We are only aware of the

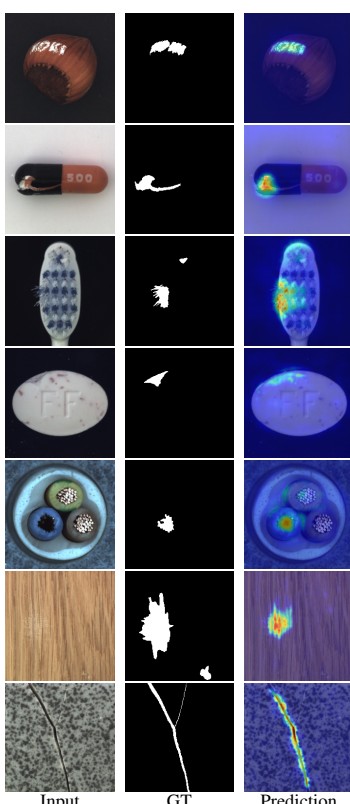

| Category | AE (SSIM) | AE (MSE) | AnoGAN | VE-VAE | MAML-AE | Ours |
|---|---|---|---|---|---|---|
| Carpet | 0.69 | 0.38 | 0.34 | 0.1 | 0.20 | 0.26 |
|  | 0.87 | 0.59 | 0.54 | 0.78 | 0.68 | 0.84 |
| Grid | 0.88 | 0.83 | 0.04 | 0.02 | 0.01 | 0.12 |
|  | 0.94 | 0.90 | 0.58 | 0.73 | 0.53 | 0.82 |
| Leather | 0.71 | 0.67 | 0.34 | 0.74 | 0.12 | 0.40 |
|  | 0.78 | 0.75 | 0.64 | 0.87 | 0.77 | 0.95 |
| Tile | 0.04 | 0.23 | 0.08 | 0.14 | 0.14 | 0.26 |
|  | 0.59 | 0.51 | 0.50 | 0.93 | 0.52 | 0.76 |
| Wood | 0.36 | 0.29 | 0.14 | 0.47 | 0.11 | 0.23 |
|  | 0.73 | 0.73 | 0.62 | 0.91 | 0.68 | 0.78 |
| Bottle | 0.15 | 0.22 | 0.05 | 0.07 | 0.02 | 0.23 |
|  | 0.93 | 0.86 | 0.86 | 0.78 | 0.56 | 0.82 |
| Cable | 0.01 | 0.05 | 0.01 | 0.18 | 0.04 | 0.18 |
|  | 0.82 | 0.86 | 0.78 | 0.90 | 0.74 | 0.80 |
| Capsule | 0.09 | 0.11 | 0.04 | 0.11 | 0.03 | 0.10 |
|  | 0.94 | 0.88 | 0.84 | 0.74 | 0.68 | 0.90 |
| Hazelnut | 0.00 | 0.41 | 0.02 | 0.44 | 0.11 | 0.40 |
|  | 0.97 | 0.95 | 0.87 | 0.98 | 0.72 | 0.94 |
| Metal nut | 0.01 | 0.26 | 0.00 | 0.49 | 0.10 | 0.28 |
|  | 0.89 | 0.86 | 0.76 | 0.94 | 0.78 | 0.78 |
| Pill | 0.07 | 0.25 | 0.17 | 0.18 | 0.10 | 0.11 |
|  | 0.91 | 0.85 | 0.87 | 0.83 | 0.62 | 0.88 |
| Screw | 0.03 | 0.34 | 0.01 | 0.17 | 0.02 | 0.08 |
|  | 0.96 | 0.96 | 0.80 | 0.97 | 0.55 | 0.86 |
| Toothbrush | 0.08 | 0.51 | 0.07 | 0.14 | 0.06 | 0.18 |
|  | 0.92 | 0.93 | 0.90 | 0.94 | 0.80 | 0.85 |
| Transistor | 0.01 | 0.22 | 0.08 | 0.30 | 0.02 | 0.18 |
|  | 0.90 | 0.86 | 0.80 | 0.93 | 0.76 | 0.80 |
| zipper | 0.10 | 0.13 | 0.01 | 0.06 | 0.04 | 0.15 |
|  | 0.88 | 0.77 | 0.78 | 0.78 | 0.68 | 0.86 |

Figure 3: Visualizations of localized anomaly by our method.

Table 1: Numerical evaluation of anomaly localization on MVTec-AD. We report both mIoU (top rows) and AUC-ROC (bottom rows) values. Col 2-5 are upper-bound methods trained with massive normal samples.

work in [32] (r-GAN) that has such adaptation capability. Specifically, the model adapts to a new scene using gradient descent with several beginning frames of a query video, after which a GAN is applied to generate future frames. Anomaly is then detected via the discrepancy between predicted future frames and the original frames. Note that the MAML-AE baseline we conducted in Section 4.1 can be seen as an ablation of r-GAN on the single-frame without temporal information.

We follow the same evaluation regime as r-GAN by training with normal samples in all 13 scenes from SH-Tech [28] and testing on UCSD Pedestrian 1, UCSD Pedestrian 2 [34], and CUHK Avenue [30]. Note that since our method is image-based, it predicts the video frames independently without leveraging any temporal information as in r-GAN. In each episode, we adapt our model with a support set containing a few normal frames randomly sampled from the target scenes. In Table 2, we compare our method against r-GAN pre-trained on SH-Tech only (r-GAN Pre-train), fine-tuned on target datasets (r-GAN Fine-tune), and with one step gradient descent with meta-learning (r-GAN MAML). We also show the performance of MAML-AE as a baseline for image-based meta-learning method. In the last section of Table 2, we present intra-dataset results as well by training with 6 scenes of SH-Tech and testing on remaining 7. We follow common evaluation protocol and measure the frame-level AUC-ROC. Without leveraging temporal information and re-training (gradient descent), our method achieves comparable results to r-GAN MAML and outperforms image-based meta-learning method by a large margin. In App B.4, we show that incorporating simple temporal information can further improve the performance.

## 4.3 Ablation Studies

**Sparse coding receptive fields.** To evaluate the effectiveness of using large receptive fields in the sparse coding layer, we conduct additional experiments on the MVTec-AD dataset, and select 5 representative categories with different levels of difficulties to present the comparisons with $l = 1$ and $l = 3$ (Sec. 3.1.1) in Table 3. Sparse coding with large receptive field clearly benefits more complex

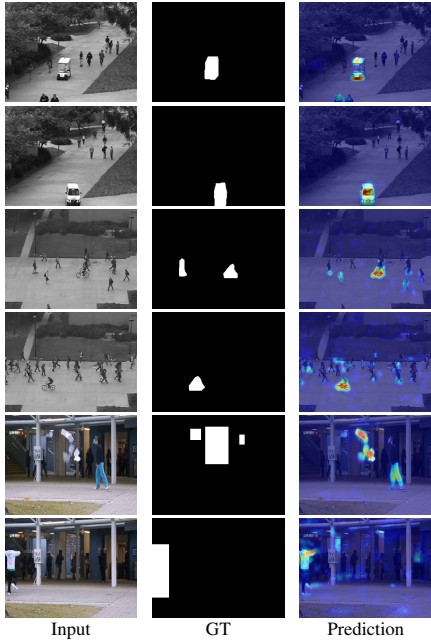

| Input | GT | Prediction |
|---|---|---|

Figure 4: Visualizations of anomaly localization with video anomaly detection.

| Target datasets | Methods | 1-shot | 5-shot | 10-shot |
|---|---|---|---|---|
| UCSD Ped 1 | r-GAN Pre-train | 73.10 | 73.10 | 73.10 |
| | r-GAN Fine-tune | 76.99 | 77.85 | 78.23 |
| | r-GAN MAML | 80.60 | 81.42 | 82.38 |
| | MAML-AE | 64.12 | 66.88 | 67.34 |
| | Ours | 77.42 | 78.12 | 78.65 |
| UCSD Ped 2 | r-GAN Pre-train | 81.95 | 81.95 | 81.95 |
| | r-GAN Fine-tune | 85.64 | 89.66 | 91.11 |
| | r-GAN MAML | 91.19 | 91.80 | 92.80 |
| | MAML-AE | 78.24 | 82.04 | 83.30 |
| | Ours | 91.22 | 92.00 | 92.45 |
| CUHK Avenue | r-GAN Pre-train | 71.43 | 71.43 | 71.43 |
| | r-GAN Fine-tune | 75.43 | 76.52 | 77.77 |
| | r-GAN MAML | 76.58 | 77.10 | 78.79 |
| | MAML-AE | 68.72 | 69.67 | 70.01 |
| | Ours | 80.68 | 83.41 | 84.46 |
| Sh-Tech | r-GAN Pre-train | 70.11 | 70.11 | 70.11 |
| | r-GAN Fine-tune | 71.61 | 70.47 | 71.59 |
| | r-GAN MAML | 74.51 | 75.28 | 77.36 |
| | MAML-AE | 66.62 | 67.12 | 68.04 |
| | Ours | 75.32 | 79.64 | 81.28 |

Table 2: Frame-level AUC-ROC for the video anomaly detection tasks.

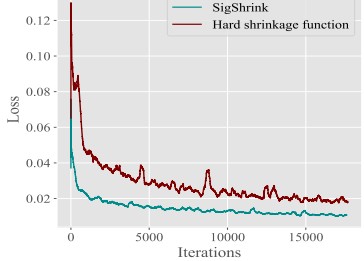

Figure 5: Loss curves with smooth (SigShrink) and non-smooth (hard-shrink RELU-like) shrinkage functions.

| Category | Leather | | Grid | | Hazelnut | | Cable | | Zipper | |
|---|---|---|---|---|---|---|---|---|---|---|
| $l = 1$ | 0.40 | 0.95 | 0.11 | 0.80 | 0.36 | 0.91 | 0.17 | 0.76 | 0.12 | 0.84 |
| $l = 3$ | 0.40 | 0.95 | 0.12 | 0.81 | 0.40 | 0.94 | 0.18 | 0.80 | 0.15 | 0.86 |

Table 3: Comparison of different sparse coding receptive fields. We report both mIoU (left) and AUC-ROC (right) values.

| Category | Leather | | | Hazelnut | | | Cable | | |
|---|---|---|---|---|---|---|---|---|---|
| Ours | 0.40 | 0.95 | 1.6e-4 | 0.40 | 0.94 | 2.4e-4 | 0.18 | 0.80 | 2.0e-4 |
| No sparsity | 0.32 | 0.90 | 0.9e-4 | 0.24 | 0.80 | 1.7e-4 | 0.12 | 0.68 | 1.5e-4 |

Table 4: Performance w/ and w/o sparsity constraint. From left to right: mIoU; AUC-ROC; the difference of averaged reconstruction errors between abnormal/normal samples.

structural objects (hazelnut, cable, and capsule), while the improvements are limited for the texture objects (leather and grid), where contextual regularization is intuitively less important.

**Shrinkage functions.** To show the benefits of smooth shrinkage function, we plot the loss curves of models trained with smooth SigShrink (Eqn. 9) and non-smooth RELU-like shrinkage (Eqn. 7) functions in Fig 5. The model with smooth shrinkage function converges notably faster in the early training stage and achieves lower loss.

**Sparsity constraint.** As discussed in Section 3.1, we impose sparsity constraint to the feature decomposition in the adaptive sparse coding layer, in order to prevent abnormal features from being well-approximated by the linear combinations of normal features, so that the reconstruction errors are effective for detecting anomaly. To validate this, we conduct experiments by removing the shrinkage function $\sigma$ in the sparse coding stage (Eqn. 6). We show comparison in Table 4 with mIoU, AUC-ROC, and the difference of averaged reconstruction errors between abnormal and normal samples. Without sparsity, the performance drops dramatically, and reconstruction errors of normal and abnormal samples become closer.

# 5 Related Work

**Anomaly detection with sparse coding.** Early efforts on adopting sparse coding in anomaly detection are based on optimization (with L1 penalty) [30, 61]. Recent advances on iterative sparse thresholding algorithms [11, 25] allow seamless integration of online sparse coding with deep neural networks, and [33] formulates the sparse coding as stack RNNs for video anomaly detection.

**Anomaly detection with generative models.** One of the core challenges in anomaly detection is that abnormal samples are usually unavailable in the training stage. Generative models are widely utilized in anomaly detection due to the capability in modeling the density of desired data distribution. Early efforts on variational autoencoders (VAE) based methods [1, 58] are arguably having hard time calibrating uncertainties in novel samples [36], accurately localizing abnormal regions through reconstruction errors [12]. Recent efforts have explored variant generative architectures like energy-based models (EBM) [20], GANs [50], and combining VAE with EBM [12]. Various methods also exploit intra-image structures [10, 7], cross-frame consistency [31], and motion-appearance consistency in videos [39] while detecting anomaly.

**Few-shot learning.** Few-shot learning is extensively explored in classification tasks. It leverages common knowledge extracted from a distribution of tasks, and induces an adaptive model that fits to a new classification task with as few as one sample per class. Proposed methods are based on optimization [18, 47, 19, 59, 46], learning metric [51, 53] and parameter prediction [22, 43, 21]. These technologies are further applied in other tasks like image generation [9, 27] and out-of-distribution detection [49].

**Energy-based models.** As a family of generative models, studies on EBMs [26] are mainly focused on probabilistic modeling and sampling over data, either unconditionally [38, 44, 40, 17, 15, 63, 57, 56, 2], or conditionally [13, 14]. This has been recently followed by extensions to other applications that include reasoning [16], latent space modeling of generative models [41], and anomaly detection [12].

# 6 Conclusion

In this paper, we introduced few-shot fast-adaptive anomaly detection. We formulated our model as an energy-based model with an adaptive sparse coding layer, of which the dictionary is directly formed by normal features of a target task. We adopted episodic meta-learning to learn common knowledge across tasks, which enables few shots adaptation. We further introduced smooth shrinkage functions, sparse coding with large receptive fields, and learning by inpainting to improve and accelerate the training. Notably, when evaluating our method's performance on industrial inspection and video anomaly detection, our method is comparable and even boasts better performance than methods trained with a large amount of normal samples. Through this work, we hope to have made a significant contribution to the important problem of anomaly detection by shedding light on our findings that anomaly detection can indeed be generalized to new tasks with a few normal samples only.

**Social Impact and Ethics.** As a general framework for few-shot anomaly detection, the proposed method does not suffer from particular ethical concerns or negative social impacts. All datasets used are public, and we have blurred all human faces in the qualitative visualizations.

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
