# Appendix

# A    Implementation Details

We adopt a residual network (ResNet) [23] based feature extractor, with ELU as the activation function. Following [15], we adopt group normalization and instance normalization for better stability of the networks. The detailed network construction is shown in Table A.

| Output size | Layers |
|---|---|
| ResNet-10 | |
| $224{\times}224{\times}3$ | Input images |
| $56{\times}56{\times}32$ | *Conv*($7{\times}7$, stride=2), *GroupNorm*, *ELU*, *AveragePool*($3{\times}3$, stride=2) |
| $56{\times}56{\times}32$ | [*Conv* ($3{\times}3$), *GroupNorm*, *ELU*] $\times$ 2, 
 *Skip connect Conv* ($1{\times}1$, stride=2), *GroupNorm* |
| $28{\times}28{\times}64$ | *Conv* ($3{\times}3$, stride=2), *GroupNorm*, *ELU*, 
 *Conv* ($3{\times}3$), *BatchNorm*, *ELU* 
 *Skip connect Conv* ($1{\times}1$, stride=2), *GroupNorm* |
| $14{\times}14{\times}128$ | *Conv* ($3{\times}3$, stride=2), *GroupNorm*, *ELU*, 
 *Conv* ($3{\times}3$), *GroupNorm*, *ELU* 
 *Skip connect Conv* ($1{\times}1$, stride=2), *GroupNorm* |
| $7{\times}7{\times}256$ | *Conv* ($3{\times}3$, stride=2), *GroupNorm*, *ELU*, 
 *Conv* ($3{\times}3$), *GroupNorm*, *ELU* 
 *Skip connect Conv* ($1{\times}1$, stride=2), *GroupNorm* |
| $7{\times}7{\times}256$ | *Conv* ($1{\times}1$, stride=2), *GroupNorm*, *Tanh* |

Table A: The architectures of feature extractor.

## A.1    Industrial Inspection

We adopt the "leave-one-out" training strategy for obtaining the results on each of the categories of MVTec-AD. All experiments are performed with the same settings and hyperparameters. We resize all images to $128 \times 128$, and do not perform any data augmentation. We adopt a simple reduced-sized ResNet as the feature extractor as shown in Table A. Following [15], we adopt group normalization (denoted as GroupNorm) instead of batch normalization, and use Exponential Linear Unit (ELU) as the activation function. We empirically observed that using Tanh as the final activation function can remarkably improve the numerical stability of the sparse coding stage as the magnitude of the feature values is effectively bounded by the final activation function.

We adopt Adam as the optimizer, with a consistent learning rate of 1e-4. We do not apply any net regularization methods like dropout or weight decay in training. The weights of reconstruction $\eta_0$ and contrastive divergence $\eta_1$ are set to 1.0 and 0.25, respectively. Each training batch contains 4 randomly sampled training tasks with 10 query ($M = 10$) for each task. All training can be conducted on a single NVIDIA Tesla A100 GPU.

We perform 8 steps of sparse coding in the adaptive sparse coding layer, with an initial $\lambda_{max} = 0.3$ and a final $\lambda_{\star} = 0.05$. We perform 5 steps of Langevin dynamics with a step size of $\beta = 1.0$ to synthesize negative samples, which we show in the examples of Figure 2 and Figure E to be sufficient for producing hard negative samples.

## A.2    Video Anomaly Detection

We resize each frame to $240 \times 320$. No data augmentation is performed. All other hyperparameters equal to those applied in industrial inspection experiments.

## A.3    MAML-AE

We adopt a full-convolutional auto-encoder network to construct the MAML-AE. A 10 layer ResNet [23] as the encoder, and consecutive transpose-convolutional layers with batch normalization and RELU activation function as the decoder to recover the feature resolution. The hyperparameters of the episodic training of MAML AE are exactly the same with those of our methods. We perform 5 steps of gradient descent as the inner-loop adaption.

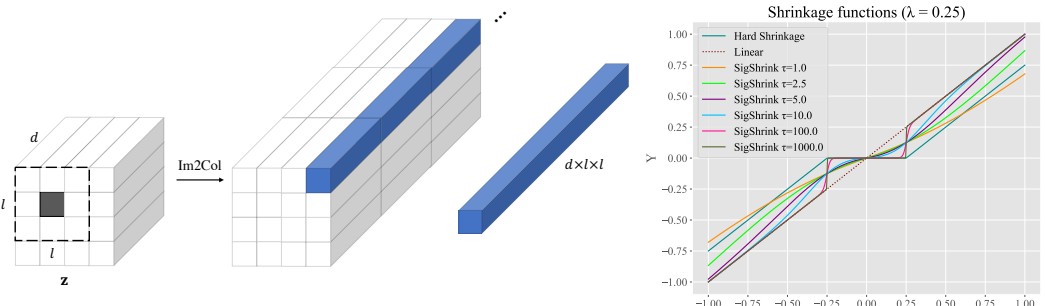

Figure A: Illustration of sparse coding layer with $l \times l$ receptive filed.

Figure B: Shrinkage functions.

We directly use the energy score in (Eqn. 8) as the anomaly score. When evaluating AUC-ROC and mIoU, we obtain the normalized energy score for each sample by uniformly normalize to the scores of all samples in test set.

### A.4 Initializing Negative Sample

In the synthesis of negative samples, we randomly place at most three random patches to each normal image. Each patch is created by one of the following:

- **Random uniform:** A random patch with the values of each pixel sampled from a uniform distribution. The minimum and maximum values of the uniform distribution is equal to those of the pixel values of the images.
- **Random consistent:** A random patch with consistent pixel values sampled from a uniform distribution. The minimum and maximum values of the uniform distribution is equal to those of the pixel values of the images.
- **Random copy-paste:** A random patch randomly cropped from the same image.
- **Random blurring:** Applying Gaussian blurring to a random patch of the normal image.

Denoting $\mathcal{U}(a, b)$ as an uniform distribution with minimum and maximum values of $a$ and $b$, respectively, the relative size of the random patch w.r.t. the original image is randomly sampled from $\mathcal{U}(0.0025, 0.025)$, and the aspect ratio is randomly sampled from $\mathcal{U}(0.01, 100.0)$. See Fig. E for illustrations of synthesized negative samples and the corresponding generative sequences of Langevin dynamics.

### A.5 Illustrations

**Sparse coding layer with receptive field.** We illustrate sparse coding with receptive field in Fig. A. An $l \times l$ sliding window is applied spatially to the feature map, and the feature volume in each window is flatten as a feature vector to perform sparse coding. In this way, the context information of each position is effectively considered in the adaptive sparse coding layer.

**Shrinkage functions.** We plot different shrinkage function in Figure B. Beside the desired property of sparse projection, the SigShrink we use enjoys the advantages of being differential and having non-zero gradients everywhere.

## B Additional ablations

### B.1 Robustness to Pose Variations

To validate the robustness of our method to pose variations, we perform additional experiments with pose variations and provide the quantitative comparisons in Table B. We report results by performing random $\pm 90^o$ rotation to samples during testing stage when adapting the trained model to novel categories. We observe that the textures categories (leather and grid) are robust to any pose variations in both the query and support samples. For the hazelnut category, where there are intrinsically significant pose variations across samples, performing any new rotations does not influence the results. On the other hand, for the pill category, where all samples are aligned horizontally (see examples in Appendix, Figure D), performing rotation to only query samples results in performance drop. Performing rotation to both query and support samples help to recover the performance. The above results suggest the robustness of our methods under the condition of sufficient sample diversity in the support set.

| Category | Leather | | Grid | | Hazelnut | | Cable | | Zipper | |
|---|---|---|---|---|---|---|---|---|---|---|
| Original | 0.40 | 0.95 | 0.12 | 0.82 | 0.40 | 0.94 | 0.18 | 0.80 | 0.15 | 0.86 |
| Rotate query | 0.39 | 0.95 | 0.12 | 0.80 | 0.40 | 0.94 | 0.17 | 0.77 | 0.13 | 0.82 |
| Rotate query and support | 0.40 | 0.95 | 0.11 | 0.82 | 0.40 | 0.94 | 0.17 | 0.78 | 0.14 | 0.85 |

Table B: Robustness to pose variations. We report both mIoU (left) and AUC-ROC (right) values.

## B.2 Comparison to Image Differencing

Image differencing and its variances are an intuitive approach for detecting image anomaly. We perform image differencing and background subtraction on the industrial inspection and video anomaly detection task, respectively, and report performance in Table C. Our method demonstrates clear advantages.

| Category | Leather | | Hazelnut | | Cable | | UCSD Ped1 | UCSD Ped2 | CUHK |
|---|---|---|---|---|---|---|---|---|---|
| Image differencing | 0.02 | 0.57 | 0.12 | 0.77 | 0.07 | 0.65 | 65.34 | 59.03 | 57.12 |
| Ours | 0.40 | 0.95 | 0.40 | 0.94 | 0.18 | 0.80 | 77.42 | 91.22 | 75.32 |

Table C: Comparing with image differencing. We report mIoU (left) and AUC-ROC (right) for industrial inspection task and AUC-ROC for video anomaly detection task.

## B.3 Robustness Against Contaminated Training Data

While it is a common practice among machine learning practitioners to assume clean training data, this may not be true in real world applications. In this section, we evaluate the robustness of our method against contaminated training data by inserting certain amount of abnormal samples. As shown in Table D, we progressively contaminate the normal training data by increasing the amount of abnormal data from 1% to 10%. Our proposed method is in general robust to data contamination, where contamination under 5% only decreases the performance slightly. The network is still able to perform decently in the extreme case of 10% contamination.

| Contamination | Leather | | Grid | | Hazelnut | | Cable | | Zipper | |
|---|---|---|---|---|---|---|---|---|---|---|
| 0% | 0.40 | 0.95 | 0.12 | 0.81 | 0.40 | 0.94 | 0.18 | 0.80 | 0.15 | 0.86 |
| 1% | 0.40 | 0.95 | 0.11 | 0.80 | 0.40 | 0.94 | 0.17 | 0.79 | 0.14 | 0.85 |
| 5% | 0.38 | 0.93 | 0.11 | 0.78 | 0.39 | 0.93 | 0.16 | 0.77 | 0.14 | 0.83 |
| 10% | 0.36 | 0.92 | 0.10 | 0.76 | 0.37 | 0.90 | 0.16 | 0.75 | 0.13 | 0.83 |

Table D: Performance evaluation against data contamination. We contaminate the normal training data by inserting increasing percentages of abnormal data.

## B.4 Leveraging Temporal Information

An EBM is agnostic to the underlying backbone network architecture, therefore it is straightforward to incorporate temporal information into the image-based anomaly detection framework described in Section 5.2. We replace the 2D convolutions in the first four layers with 3D convolutions, and have the model accepts a 5-frame input instead. Without heavy tuning, we report 5-shot performance on both '1 frame' (original, without temporal) and '5 frames' (with temporal) in Table E. The results show that when temporal information is added, it leads to further improvements.

## B.5 Adaptive Sparse Coding Layer with Auto-encoder

Auto-encoders are widely used in anomaly detection with the localization performed by measuring the reconstruction error. An straightforward baseline of few-shot anomaly detection can be constructed by plugging the proposed adaptive sparse coding layer as the bottleneck of an autoencoder. However, we empirically find it very difficult to train a decoder that can generalize well to unseen classes. As visualized in Fig. C, the poor generalization of the decoder consistently results in drastically high reconstruction error when applied to novel test tasks. Quantitatively, this simple baseline consistently delivers nearly 0 mIoU, therefore is not included in the comparisons. On the other hand, utilizing only an encoder and using the backward gradient of the EBM to localize the anomaly demonstrate outstanding robustness on novel tasks that are unseen during the training stage as we discuss in Sec. 4.

| Category | UCSD Ped1 | UCSD Ped2 | CUHK | Sh-Tech |
|----------|-----------|-----------|------|---------|
| 1 frame | 78.12 | 92.00 | 83.41 | 79.64 |
| 5 frames | 79.06 | 91.94 | 84.27 | 80.80 |

Table E: Performance comparisons with temporal information incorporated.

Training

Testing

| Support | Query | Reconstructed query | | Query | Reconstructed query | Reconstruction error | Ground-truth anomaly |

Figure C: Applying the proposed adaptive sparse coding layer to an auto-encoder for few-shot anomaly detection and localization suffers severely from the poor generalization of the decoder, which delivers consistently high reconstruction error on novel tasks.

## C  Qualitative Results

We present in Figure D additional anomaly localization results of categories with different anomalies in the MVTec-AD dataset.

## D  Additional Figures

Common practice of sampling from EBMs is computationally costly, requiring as many as 50 steps of Langevin Dynamics when initialized with full noise, instead of a synthesized negative sample as shown in Fig. E.

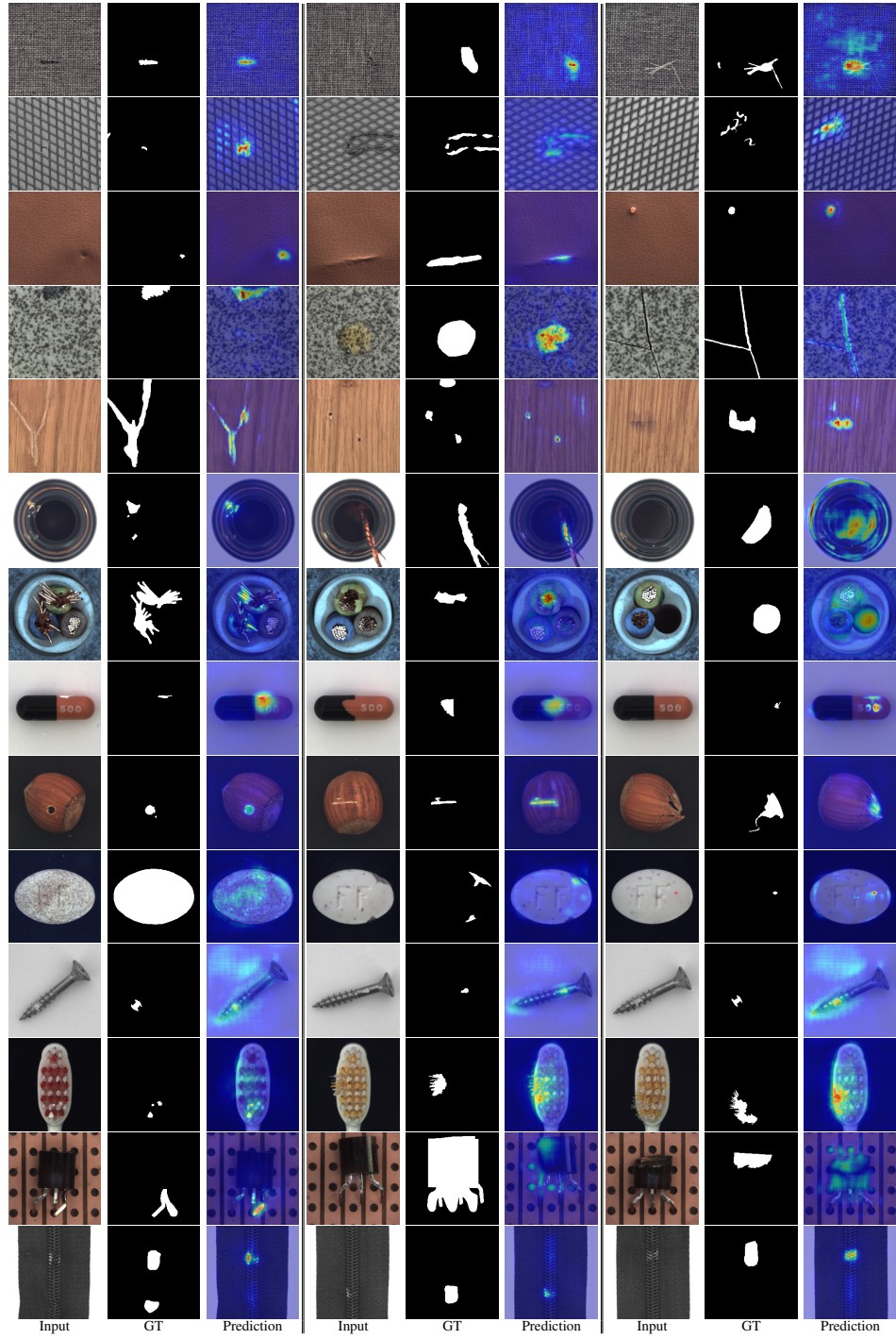

| Input | GT | Prediction | Input | GT | Prediction | Input | GT | Prediction |

Figure D: Visualizations of anomaly localization on industrial inspection data. All results are obtained by adapting the model using 10 normal samples only.

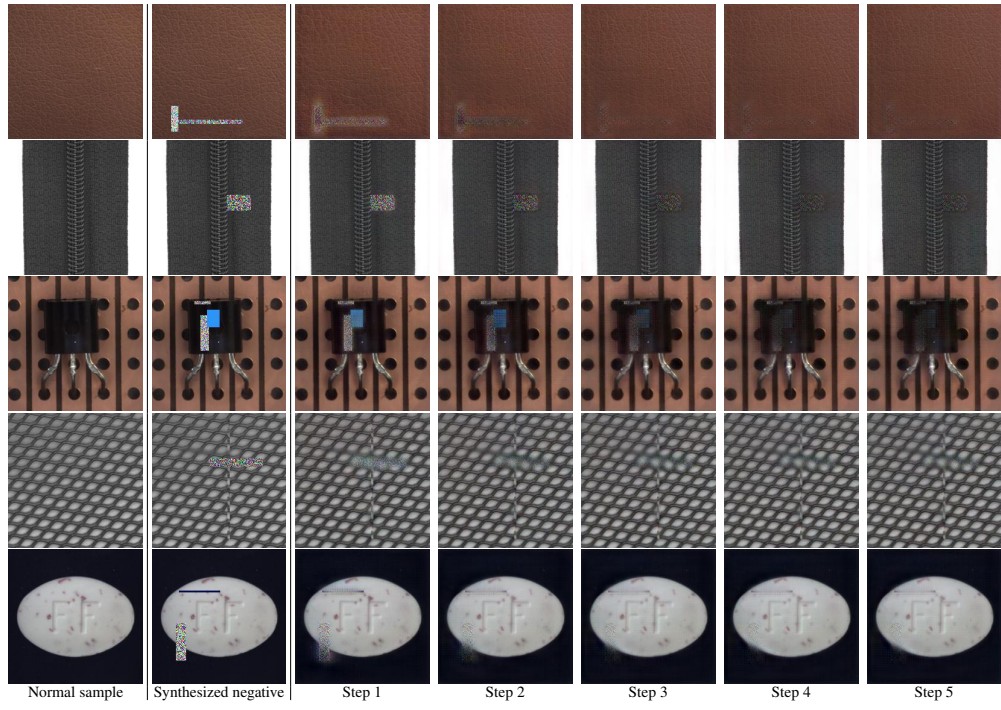

Figure E: Sampling outputs after a few steps of Langevin Dynamics starting from each synthesized negative sample. 5 steps of Langevin dynamics are sufficient to quickly generate hard negative samples with minor artifacts.