# OpenReview forum: "Few-Shot Fast-Adaptive Anomaly Detection"
_NeurIPS.cc/2022/Conference — NeurIPS 2022 Accept_

### Official Review · Reviewer_3BqJ · 2022-07-11

**Rating:** 7
**Confidence:** 3
**Soundness:** 2 fair
**Presentation:** 2 fair
**Contribution:** 2 fair

**Summary:**

This work presents a new algorithm for image-based anomaly detection. The algorithm relies upon the contrastive-divergence method for training energy based models and it presents a differentiable "sparse-coding layer" which seems to be novel. Beyond the algorithmic innovations the paper also presents practical techniques for improving the speed of generating samples from an EBM for images. This work also presents experiments for few-shot learning their proposed algorithm across new domains.

The basic idea is to use sparse-coding as a way of approximating whether a novel sample is similar to in-distribution samples or not. The innovation lies in implementing sparse-coding in a differentiable manner that lets them train the entire architecture in an end-to-end manner.

**Questions:**

Please see the questions mentioned in the "Strengths and Weaknesses" section.

**Limitations:**

Yes the authors adequately discusses the limitations and potential negative societal impact of their work.

**Strengths And Weaknesses:**

This paper presents a new algorithm for image anomaly detection with good empirical performance and some innovative ideas. I believe the results/techniques could be quite useful in other contexts where researchers want to perform nearest-neighbor decomposition in a differentiable manner.

However the techniques in the paper sometimes seem to be put together a little haphazardly and the paper does not do a good job of clearly explaining the contribution of many of the important parts. For example, a lot of space is devoted in the paper on episodic training as a way of enabling few-shot learning. However it isn't clear from the experiments that how important is the episodic meta-learning for few-shot learning. The biigest help for few-shot learning comes from the sparse-coding layer which doesn't have any learnable parameters.  For example if instead of breaking up the training into $I$ separate episodes where each episode was one task, if we simply trained in one episode on all the data what will be the performance loss ?

Also the paper sometimes makes claims that are a hard to understand/verify. For example, the paper claims that proposed method is the "first to allow [adaptation] to new tasks with a single forward pass without any training". However even a simple process such as finding the distance to the 1-nearest neighbor allows some level of adaptation in a single forward pass.

--- After the author response --

the authors replied back with more evidence about the importance of meta-learning which seems sufficient, updating my rating now

---

> ### Author Response · Authors · 2022-08-01
> **Response to Reviewer 3BqJ**
>
> We thank the reviewer for their constructive feedback! We have addressed the comments below.
>
> **1. Contribution**
>
>
> *‘For example if instead of breaking up the training into separate episodes where each episode was one task, if we simply trained in one episode on all the data what will be the performance loss ?’*
>
> Episodic training is widely used to empower a meta-learner with the ability of solving novel tasks through learning the adaptation from existing tasks to a new task. The episodic training mimics the testing scenario, by first showing the model with normal samples of a new tasks, and then update the model based on the evaluation on the testing samples from this task.
> In real-world application, we usually care about one task at a time. Mixing all data in one episode without learning how to adapt the train-test discrepancy will hurt the generalization ability at test time.
> In our observation, this will bring the model performance to nearly random guess (with AUC-ROC close to 0.5).
>
>
> Episodic training is a standard way of training models to perform few-shot adaptation, while different works might propose distinct adaptation methods. In our paper, we propose adaptive sparse coding layer to perform the model adaptation, which demonstrates both better efficiency and effectiveness.
>
>
> In summary, The failure of the suggested mixed-task experiment reveals that episodic training is indispensable to learn the adaptation, while the proposed model can adapt itself to a novel task thanks to the adaptive sparse coding layer.
>
>
>
>
>
> **2. Claims**
>
> The claim "first to allow [adaptation] to new tasks with a single forward pass without any training" we made in the original manuscript is mainly based on the comparisons with state-of-the-art anomaly detection method published in recent years, especially the ones we compare against in Section 4.
>
> Simple methods such as nearest neighbor and image differencing (Appendix Section B.2) can also adapt without training, however their performance is far from satisfying. For example, according to Appendix Table C, the performance of image differencing on some challenging categories degrades to nearly random guess (AUC-ROC close to 0.5). We appreciate that the reviewer point this out, and we have modified the claim into "first to allow [adaptation] to new tasks with a single forward pass without any training meanwhile demonstrate strong performance on challenging anomaly detection and localization tasks", and marked it in blue in Section 4.1 of the revised manuscript.

---

> > ### Comment · Reviewer_3BqJ · 2022-08-05
> > **reply to the authors.**
> >
> > Thanks for the updates. I have updated my score based on the author response.

---

> > > ### Author Response · Authors · 2022-08-05
> > > **Reply to Reviewer 3BqJ**
> > >
> > > We appreciate reviewer's prompt feedback, and feel free to let us know if there are further questions.

---

### Official Review · Reviewer_ezfd · 2022-07-11

**Rating:** 8
**Confidence:** 4
**Soundness:** 3 good
**Presentation:** 3 good
**Contribution:** 3 good

**Summary:**

The paper proposes a framework for building an anomaly detector which can be adapted to new tasks at inference time with a few examples from the new task. This is in-contrast to standard anomaly detection methods which had to be rebuild for each new task. For example, a standard autoencoder can be trained for normal tasks(like video surveillance)  and anything with a large reconstruction error can be labeled as anomalous. Now if the scene or camera changes then the autoencoder will have to be re-trained. This paper proposes a three tier framework where the trained anomaly detector can be adapted at inference time.

**Questions:**

How well does the method do with so called "co-variate shift". The video surveillance example might be an example of co-variate shift ?

**Limitations:**

No potential negative societal impact.

**Strengths And Weaknesses:**

Strengths:

(i) Paper clearly defines the three tasks G1, G2 and G3 which makes it easy to comprehend.

(ii) The idea of using intermediate examples during the LD process as anomalies and using them to train an EBM is novel. Similarly the idea of borrowing ideas from sparse autoencoders and "re-training" it with a few examples makes the detector adaptive.

(iii) The results are quite impressive. For example, in Table 1, the proposed method is even competitive with "upper-bound" where the autoencoders (or other methods) are trained with many samples from each category.

Weakness:

(i) My slight concern is that the whole framework might be too complicated to be actually deployed.

---

> ### Author Response · Authors · 2022-08-01
> **Response to Reviewer ezfd**
>
> Thanks for reviewer's valuable comments!
>
> **1. Algorithm deployment**
>
> The presentation of the whole workflow might seem heavy, the real-world deployment, which only includes new-task adaptation and inference, can be simple.
> To show this, we have added one more algorithm box showing the inference steps in revised Appendix Section A.6, Algorithm 2.
> Even compared to few-shot anomaly detection method r-GAN, our method benefits from better efficiency and simplicity as the adaptation does not involve any changes to the meta-learned parameters in the feature extractor. We believe this can be an advantage while deploying the algorithm on edge devices.
>
> **2. Co-variate shift**
>
> As visualized in Figure 3 and Figure 4, both video surveillance datasets and industry inspection dataset we adopted exhibits noticeable variance among tasks.
> The strong performance of the cross-dataset video surveillance anomaly detection, and the image anomaly detection with distinct object appearance demonstrate the robustness of the proposed method against co-variate shift.

---

> > ### Comment · Reviewer_ezfd · 2022-08-07
> > **Update**
> >
> > Thanks for updating the algorithm and clarification on co-variate shift

---

### Official Review · Reviewer_hTet · 2022-07-12

**Rating:** 7
**Confidence:** 2
**Soundness:** 3 good
**Presentation:** 2 fair
**Contribution:** 3 good

**Summary:**

The paper presents an anomaly detection technique based on EBM (energy based model). The proposed technique is supposed to help adapt to new tasks faster with just a few rounds of training.

**Questions:**

Could the authors elaborate more clearly on how the task-specific dictionary is created?

**Strengths And Weaknesses:**

The proposed algorithm is a combination of several existing techniques. A fundamental contribution is hard to identify.


The paper is very hard to read and also has numerous typos such as:
Line 11: associates => associate
Line 47: learn => learns
Line 146: pixe => pixel
Line 178: SigShink => SigShrink
Line 245: auto-encode => auto-encoder
Line 249, 251: adapt => adapting


Figure 1 does not really have labeled parts a, b. It also is not very intuitive or informative in explaining parts of the algorithm, e.g., line 135 refers to Fig (a) in relation to the dictionary D. But no additional clarity about D is offered by reference to this figure.


The construction of task-specific dictionary D has not been explained very well. In general it would help to present the content in Sections 3.1 and 3.2 in a more intuitive manner.


It certainly helps that the paper does an ablation study (Sec 4.3) and shows the benefits of the various parts of the algorithm.

=============
Update after author rebuttal:

Thanks to the authors for addressing my concerns. I am revising my scores.
--------------------

---

> ### Author Response · Authors · 2022-08-01
> **Response to Reviewer hTet**
>
> Thanks for the constructive feedback! We have addressed reviewer's comments and revised the manuscript accordingly.
>
> **1. Figure 1**
>
> In the original manuscript, we labeled parts (a) and (b) in the figure besides the name of each module.
> And the dictionary $\mathbf{D}$ was marked inside the green box in Figure 1 (a).
> We appreciate reviewer's suggestion. We changed the font size in Figure 1 and marked the labels with different colors in the revised version to make them more visible.
>
> **2. Construction of task specific dictionary $\mathbf{D}$**
>
> We discussed the construction of the task-specific dictionary in Section 3.1 and visualized it in Figure 1 (a).
> To further improve the presentation, we have added Algorithm 2 in Appendix to demonstrate how exactly the proposed framework works on a task at test time.
> Specifically, for a testing task indexed by $i$,
>
> - $K$ normal samples are first fed into the feature extractor, which outputs the normal feature $\mathbf{Z}^i \in \mathbb{R}^{K \times d \times h^\prime \times w^\prime}$, where $h^\prime$ $w^\prime$ correspond to the spatial dimension, and $d$ is the channel dimension.
>
> - The normal features are then reshaped in to $\mathbf{D}^i \in \mathbb{R}^{d \times K h^\prime  w^\prime}$ to construct the dictionary. Note that the reshape step will not introduce additional parameters.
>
> The task-specific dictionary now contains $K h^\prime w^\prime$ dictionary atoms, each of which has $d$ dimensions and corresponds to a image patch of a normal sample. Please kindly let us know and share your valuable comments regarding further improvements.
>
>
> **3. Typos**
>
> Thanks for pointing them out, we have fixed typos in the revised manuscript.

---

### Meta-Review · Area_Chair_uNED · 2022-08-23

**Recommendation:** Accept
**Confidence:** Certain

**Metareview:**

The final consensus from three reviewers knowledgeable in the field was that the paper makes an interesting contribution in the area of anomaly detection. The empirical results were seen as particularly impressive, and the treatment of intermediate samples from Langevin dynamics as abnormal was also seen as offering some novelty. My own assessment is a bit more qualified than the reviewers: while the empirical results are certainly nice, the approach itself seems a somewhat complex combination of ideas from the literature. (This complexity was also noted in one of the reviews.) While we uphold the reviewers' verdict, we encourage the authors to spend a bit more time (perhaps in Sec 2) drawing out some higher-level insights of the proposed approach, and whether there might be simpler alternatives that could also work well.

**Award:**

No

---

### Decision · Program_Chairs · 2022-09-14

Accept